# Ultrasound Control of Cervical Regeneration after Large Loop Excision of the Transformation Zone: Results of an Innovative Measurement Technique

**DOI:** 10.3390/diagnostics13040791

**Published:** 2023-02-20

**Authors:** Vincenzo Pinto, Miriam Dellino, Carla Mariaflavia Santarsiero, Gennaro Cormio, Vera Loizzi, Valentina Griseta, Antonella Vimercati, Gerardo Cazzato, Eliano Cascardi, Ettore Cicinelli

**Affiliations:** 1Department of Interdisciplinary Medicine (DIM), University of Bari “Aldo Moro”, Piazza Giulio Cesare 11, 70124 Bari, Italy; 2Gynecologic Oncology IRCCS Istituto Tumori “Giovanni Paolo II” Bari, 70124 Bari, Italy; 3Department of Interdiscipliniary Medicine (DIM), University of Bari, 70121 Bari, Italy; 4Department of Precision and Regenerative Medicine and Ionian Area, University of Bari “Aldo Moro”, Piazza Giulio Cesare 11, 70124 Bari, Italy; 5Section of Molecular Pathology, Department of Precision and Regenerative Medicine and Ionian, Area (DiMePRe-J), School of Medicine, Aldo Moro University of Bari, 70100 Bari, Italy; 6Department of Medical Sciences, University of Turin, 10124 Turin, Italy; 7Pathology Unit, FPO-IRCCS Candiolo Cancer Institute, 10060 Candiolo, Italy

**Keywords:** 3D ultrasound, cervical regeneration, cervical intraepithelial neoplasia, large loop excision of the transformation zone, LEEP, LLETZ

## Abstract

The objective of this research is to evaluate cervical regeneration after large loop excision of the transformation zone (LLETZ) through the identification of a new sonographic reference point at the level of the uterine margins. In the period March 2021–January 2022, a total of 42 patients affected by CIN 2–3 were treated with LLETZ at the University Hospital of Bari (Italy). Before performing LLETZ, cervical length and volume were measured with trans-vaginal 3D ultrasound. From the multiplanar images, the cervical volume was obtained using the Virtual Organ Computer-aided AnaLysis (VOCAL™) program with manual contour mode. The line that connects the points where the common trunk of the uterine arteries reaches the uterus splitting into the ascending major branch and the cervical branch was considered as the upper limit of the cervical canal. From the acquired 3D volume, the length and the volume of the cervix were measured between this line and the external uterine os. Immediately after LLETZ, the removed cone was measured using Vernier’s caliper, and before fixation in formalin, the volume of the excised tissue was evaluated by the fluid displacement technique based on the Archimedes principle. The proportion of excised cervical volume was 25.50 ± 17.43%. The volume and the height of the excised cone were 1.61 ± 0.82 mL and 9.65 ± 2.49 mm corresponding to 14.74 ± 11.91% and 36.26 ± 15.49% of baseline values, respectively. The volume and length of the residual cervix were also assessed using 3D ultrasound up to the sixth month after excision. At 6 weeks, about 50% of cases reported an unchanged or lower cervical volume compared to the baseline pre-LLETZ values. The average percentage of volume regeneration in examined patients was equal to 9.77 ± 55.33%. In the same period, the cervical length regeneration rate was 69.41 ± 14.8%. Three months after LLETZ, a volume regeneration rate of 41.36 ± 28.31% was found. For the length, an average regeneration rate of 82.48 ± 15.25% was calculated. Finally, at 6 months, the percentage of regeneration of the excised volume was 90.99 ± 34.91%. The regrowth percentage of the cervical length was 91.07 ± 8.03%. The cervix measurement technique that we have proposed has the advantage of identifying an unequivocal reference point in 3D cervical measurement. Ultrasound 3D evaluation could be useful in the clinical practice to evaluate the cervical tissue deficit and express the “potential of cervical regeneration” as well as provide the surgeon useful information about the cervical length.

## 1. Introduction

Cervical intraepithelial neoplasia (CIN) represents a precancerous lesion in the cervical epithelium [1]. Regression in CIN I is highly likely (60–70% in 12 months) and a follow-up, usually on annual basis with pap test and HPV test, is indicated [2]. On the contrary, in CIN II and III (CIN2+), surgical treatment is recommended. For these lesions, the American Society for Colposcopy and Cervical Pathology guidelines [3] and the European Guidelines for Quality Assurance in Cervical Cancer Screening [4] recommend excisional treatment as the primary therapeutic strategy, especially large loop excision of the transformation zone (LLETZ). This procedure is performed as an outpatient under local anesthesia, and it allows a thorough histological evaluation of the cervical sample and its margins [5,6].

The great majority of women requiring LLETZ are of childbearing age and wish to preserve their reproductive potential. In these women, excisional treatment could determine an increased risk of preterm delivery (PTD) [7]. In detail, the obstetric risk appears to be related with the length of the cone and the size of the excised specimen [8], but confidence intervals are wide, and it is still unclear whether a safe depth exists [9]. According to a multicentric case-control study, an excision deeper than 15 mm may be considered as a risk factor for PTD (relative risk 2.04 confidence interval 95% 1.41 to 2.96) [9]. The transverse diameter of the excised specimen does not seem to be associated with a lower gestational age at birth [10].

Several studies reported progressive cervical tissue regeneration following the excisional procedure by the sixth post-operative month [11,12,13,14,15]. The grade of CIN was I–III in four studies [11,12,14,15], Chikazawa et al. examined only CIN III [13]. In all studies, patients underwent LLETZ, Ciavattini et al. [15] including also carbon dioxide laser conizations. In one study [11], cervical regeneration considered the measurement of the mean crater size. In the remaining investigations, the differences of cervical length and/or volume, before and after treatment, were considered. Chikazawa et al. reported a mean increase in the cervical length of 5.23 mm in the period between 2 and 6 months after LLETZ [13]. After 6 months from excision, Nicolas observed a cervical length regrowth of 71%, Ciavattini et al. observed a mean cervical length regrowth of 89% and 86% for length and volume, respectively [15]. In the same time frame, Papoutsis et al. reported a volume regeneration of 81% and a length regeneration of 78% [14]. The described anatomical recovery may reduce the risk of negative obstetric outcome after LLETZ [16]. Most data about cervical regeneration are retrospective, originate from cohort or linkage studies and are conditioned by many confounding factors such as the calculation of the absolute volume of the cone or the proportional volume of excised tissue [17,18,19]. Furthermore, in every 2D and 3D sonographic study, the measurements of cervical length and volume reveal evident limits. Cervical length corresponds to the length of the cervical canal. It is anatomically measured by taking the internal and the external uterine os as limits. However, unlike what happens in pregnant women, the sonographic identification of the internal uterine os appears very challenging because it does not rely on a precise reference point. Measurements thus are open to interpretative subjective errors, limiting the comparison between studies [6].

We conducted a single center, prospective, observational study on patients with CIN 2+ treated with LLETZ in order to define the percentage of cervical regeneration focusing our attention on the volume and the length of the cervix.

The aim of our study is to analyze, through the use of three-dimensional ultrasound, the changes in cervical length and volume after LLETZ. Compared to other previous studies, we considered as an upper limit a new unequivocal reference point that would allow us to reproduce the measurements in prospective evaluations.

## 2. Materials and Methods

### 2.1. Inclusion and Exclusion Criteria

All patient affected by CIN 2-3 and treated with LLETZ at the University Hospital of Bari (Italy) in the period March 2021–January 2022 were included in the study. Last follow-up was completed on July 2022. Cervical intraepithelial neoplasia grade 2–3 was detected by pathologists, on colposcopy-guided cervical biopsies, using the 5th WHO Classification 2020 [20]. Patients not available to respect the subsequent follow-ups (three sonographies at 6 weeks, 3 and 6 months after LLETZ) were excluded from the study along with patients with previous cervical excisional treatments, with diabetes or autoimmune diseases as lupus erythematosus systemic, rheumatoid arthritis and systemic sclerosis. Each patient was informed about the procedures and gave their informed consent to allow data collection for research purposes. Considering that data analyzed in this study were collected during routine clinical activity and fully anonymized, and that investigators did not perform any interventional procedure, formal Institutional Review Board approval was not required due to the observational nature of the study. It was not advertised, and no remuneration was offered to the patients to enter or continue the study.

### 2.2. Pre-Excisional Cervical Examination

Cervical length and volume were measured with trans-vaginal 3D-ultrasound before performing LLETZ. Specifically, sonographies and off-line analyses of 3D volumes were carried out by a sonologist (C.M.S.), other than the surgeon, using a Samsung WS80A (Samsung, Seoul, Republic of Korea) ultrasound system with a 5–9 MHz endo-cavitary volumetric probe. The 3D volume of the whole uterine cervix was acquired in accordance with the technique described by Kim et al. [21]. In detail, a sagittal plane of the cervix showing the external os; the cervical canal and continuing with the lower endometrium was identified. The image was then magnified until the cervix occupied 75% of the monitor screen. Three-dimensional volumetric fast acquisition was set, with a sweep angle of 90°. From the multiplanar images, the cervical volume was obtained using the Virtual Organ Computer-aided AnaLysis (VOCAL™) program with manual contour mode. The lower uterine segment, the vaginal wall, and the uterine arteries were not included when the contours were drawn. Once the outline of the cervix was traced manually, the VOCAL™ program computed automatically the volume of the cervix, which was expressed in ml [21]. In an earlier study [22], the measurement of uterine volume with VOCAL™ was found to be very accurate. A coefficient of correlation = 0.97 was demonstrated for the comparison between the estimated uterine volumes obtained by this technique and uterine dry weight [22].

Unlike Kim’s and other previous studies [12,16,21,22], we considered a different upper limit of the cervical canal in our measurements. Because there is no standardized ultrasound procedure to evaluate the cervix in non-pregnant women, internal uterine OS is difficult to detect, and cervicometry may have different ultrasound-dependent measurements. This limit was identified in the line that connects the points where the common trunk of the uterine arteries reaches the uterus splitting into the ascending major branch and the cervical branch. It is clearly visible by ultrasound, with the help of color Doppler (Figure 1).

Therefore, in this study, from the acquired 3D volume, the length and the volume of the cervix were measured between this line and the external uterine os (Figure 2).

Because the cervix has a shape resembling a cylinder, measurements were also performed using an alternative method to the VOCAL™ contour mode. Specifically, the data obtained on the multiplanar images were entered into the geometric formula of cylinder as by Song and Papoutsis [14,18]: volume = π × L × [$\frac{\left(D1\right)+\left(D2\right)}{4}]$ × 2. In more detail, the distance between the external os and the aforementioned line drawn between the points where the common trunk of the uterine arteries reaches the uterus was defined as length (L); the center of this line was considered the midpoint of the cervical canal and was used to draw the transverse (D1) and the anteroposterior (D2) diameter of the cervix. In order to perform a safer excision, before LLETZ, cervicometry (length and volume) was always communicated to the surgeon.

### 2.3. Cervical Excisional Procedure

After the preoperative ultrasound cervical study and the execution of local anesthesia, LLETZ was performed as an outpatient procedure. To avoid any error in measuring the excised specimen, we made sure that local anesthetic (1–1.5 mL of mepivacaine hydrochloride 20 mg/mL) was injected outside the presumed cut line. All treatments were executed by one surgeon (V.P.) under strict colposcopic guidance with rounded loops (width 25–30 mm) selected according to the tissue area to remove. In women with type 3 transformation zone and in suspected glandular lesions, immediately after LLETZ excision, a second deeper excision was associated with a square loop of 10 mm width × 8 mm height (“top hat procedure”). The whole wound surface never underwent systematic ball diathermy. When needed, only selective coagulation for hemostasis control was performed.

### 2.4. Cone Measurements

The removed cone was immediately measured by a surgeon, using Vernier’s caliper, to acquire its three dimensions (expressed in mm). Subsequently, before fixation in formalin, the volume of the removed tissue (expressed in ml) was measured by the fluid displacement technique based on the Archimedes principle. In detail, the cone was submerged in a fluid-filled tube, and the variance in fluid levels was assessed [23]. This step is important because, as the literature reports, the measurement carried out by the pathologists after formalin fixation is about 7% lower than in the macroscopic analysis executed immediately after LLETZ [16].

### 2.5. Follow-Up

After LLETZ, patients expressed their availability for serial follow-up at 6 weeks, 3 months, and 6 months. The volume and length of the residual cervix were also assessed using 3D ultrasound, with off-line processing of the volumes stored at sonography. We evaluated two different parameters indicating cervical regrowth after LLETZ: cervical regeneration compared to the removed cone size (tissue deficit regeneration) and cervical regeneration compared to cervical biometry before excision. The tissue deficit regeneration at the site of the cervical crater was described according to Papoutsis’ formula [19]: % regeneration = [(a − b)/a] × 100 (a = cervical tissue deficit immediately after LLETZ; b = tissue deficit at follow-up (cervical volume before LLETZ—cervical volume measured at follow-up). The initial tissue deficit represents the size of the cone, while the tissue deficit regeneration at follow-up is the cervical volume variation measured before LLETZ and at follow-up. We also evaluated the cervical regeneration proportion (%) of both volume and length compared to biometry before LLETZ, not considering the cone size and in accordance with the formulas used by Song et al. [18] (volume regeneration proportion = postoperative cervical volume/pre-operative cervical volume; length regeneration proportion = postoperative cervical length/pre-operative cervical length). As described for pre-excisional evaluations, follow-up sonographies were performed by a single sonologist. To avoid any form of conditioning, in follow-up sonographies performed after LLETZ, the sonologist was not aware about the volume of the cervical excision.

### 2.6. Statistical Analysis

Statistical analyses were performed using Excel software-version 2202 (Microsoft, U.S.A.). The agreement in the measurement of cervical volume between VOCAL™ contour mode and the geometric formula of the cylinder was evaluated with the Bland–Altman analysis.

## 3. Results

In the period between March 2021 and January 2022, 62 non-pregnant patients aged 26–61 years (median 33.5 years) who underwent LLETZ for CIN2+ were enrolled after acquiring written consent. Out of the 62 initially included patients, 4 repeated LLETZ, 1 underwent hysterectomy for a glandular lesion, 11 were lost to follow-up or did not respect the scheduled follow-up, and 4 were excluded because one and/or both the uterine arteries were not identified at ultrasound. Forty-two women completed the controls up to the sixth month (Figure 3). Thirty-one (73.80%) were nulliparous, and 5 (11.90%) were post-menopausal. Prior to LLETZ, the volume and the length of the cervix measured 13.53 ± 6.07 mL and 28.42 ± 5.11 mm, respectively. The volume and the height of the excised cone were 1.61 ± 0.82 mL and 9.65 ± 2.49 mm. These values corresponded to 14.74 ± 11.91% and 36.26 ± 15.49% of baseline values, respectively. At 6 weeks, there was considerable heterogeneity in the variation of the measurements between the studied patients. In fact, about 50% of cases reported an unchanged or lower cervical volume compared to the baseline pre-LLETZ values. In the remaining patients, a slight regeneration was found as early as 6 weeks after the procedure. The average percentage of volume regeneration in examined patients is equal to 9.77 ± 55.33% (min. -43.05; max. 67.31; median 8.87; Q1 = −16.57; Q3 = 21.88). In the same period, cervical length regeneration rate was 69.41 ± 14.8%. The same evaluations were repeated 3 months after LLETZ. The results about the volume and the length of the regenerated tissue and the percentage of regeneration at 6 months are reported in Figure 4.

Regarding the percentage of volume and length recovered at 6 weeks, 3 months and 6 months, compared to pre-LLETZ cervical biometrics, the data are reported in Figure 5.

When compared, the VOCAL™ contour mode technique and the geometric formula of the cylinder using the dataset obtained by a 3D scan showed a good agreement (mean percentage of difference = 2.71 at Bland–Altman plot (Figure 6). 

## 4. Discussion

In the treatment of CIN 2+ lesions, LLETZ may assure the right balance between oncological safety (cure rates up to 90%) and conservative management. For this purpose, cervical excision treatments should be performed under colposcopy guidance, possibly after an accurate ultrasound evaluation, in order to reduce the amount of healthy tissue removed and, at the same time, to ensure excisional accuracy [24]. The surgeon performing LLETZ can define the depth of the excision. They cannot settle the percentage of removed cone compared to the starting cervical volume unless a preventive biometric evaluation of the uterine cervix is performed. Many studies have therefore considered sonography to evaluate the length [12,13,25] and in some cases the volume [6,14,15,18] of the cervix before and after LLETZ. When cervical regeneration post-LLETZ was considered, the results were overlapping. In a time frame of 6 months after surgery, cervical regeneration was not less than 70%. Papoutis et al. correlated the tissue deficit at 6 months with the proportion of tissue excised [19]. They demonstrated that the regenerative process is above 75% only when the volume of the cone excised is below 14% of baseline cervical volume [19]. Cervical length has always been measured by limiting the internal and external uterine os. However, unlike the pregnant uterus, the internal uterine os is difficult to detect, and cervicometry shows a wide intrapersonal variability. The absence of a standardized sonographic procedure to evaluate the cervix does not facilitate either the comparison between different studies [6] or any longitudinal evaluation. Founta et al. reported the use of Magnetic Resonance Imaging for cervical measurement [17], which is a technique suitable for experimental purposes only and not for routine clinical practice due to high costs and long execution time. Dückelmann et al. [16] identified at ultrasound the internal uterine os on the plane perpendicular to the cervical canal at the lower limit of the endometrial line. Since the aim of our study is to analyze cervical length and volume variations over time and not their absolute value, we considered, as an upper limit, an unequivocal reference point that would make it possible to reproduce the measurements in prospective evaluations. In our study, this reference point was identified as the line that connects the points where the common trunk of the uterine arteries reaches the uterine lateral margins and splits into the ascending major branch and the cervical branch. Although this measurement is not anatomically flawless and leads to a slight overestimation of both the longitudinal diameter and the volume of the cervix, we have standardized our criterion for identifying the upper limit of our measurements. Through the systematic application of this measurement technique on our sample, we confirmed the gradual post-excision cervical regeneration over 6 months. Moreover, unlike other previous studies that established follow-up after 6 months [6,14], we opted for the repetition of a first follow-up ultrasound at 6 weeks and 3 months after surgery, allowing us to monitor not only the presence and extent of regeneration but also its progress over time. In addition, the calculation of the percentages of cervical regeneration with respect to the values of the excised cone (as per the study by Papoutsis et al. [19]) and to the basal cervical biometric values (as per the study by Song et al. [18]) allowed us to compare the two methods (Table 1).

In our study, we demonstrated a consistent cervical regeneration over time with both methods. The results compared to baseline values show percentages of regeneration of 88% after only 6 weeks, up to 97% after 6 months. However**,** these data must be carefully evaluated, since LLETZ determines an excision of no more than 10–20% of the cervix, and therefore, cervical regrowth cannot be worse than 80%. In our opinion, cervical regeneration measurement in consideration of the removed cone size may be more useful in these studies, since it makes it possible to correlate even very different excisions.

The percentages of cervical volume regeneration have a far greater standard deviation compared to length. Therefore, in the evaluation of regeneration, the measurement of cervical length can offer more accurate information and less susceptibility of variation when compared to cervical volume. Comparing the regeneration of both the volume and the length, the cervical length regenerates faster than volume, but this difference tends to decrease over time. Indeed, at the end of the 6th month of follow-up (which previous studies have defined as the conclusion of the regenerative process [18,25]), both the volume and the cervical length are similarly regenerated. We measured a 90.99% for the volume and 91.07% for the length compared to the excised cone size. The slight difference in the dynamics of cervical regeneration recorded in the first 6 weeks, which seems to be faster in height than volume, suggests an early remodeling mechanism rather than a cervical regrowth. By the way, after cervical excision, the literature reports a regenerative process that originates from the repair cells of epithelial and stromal tissue and adapts to the new anatomy, representing a mechanism of cervical remodeling [15]. Further investigations are needed to explain this observation. The nearly complete cervical regeneration, however, does not explain the increased risk of PTD in women who have undergone LLETZ. Phadnis et al. [23] supposed possible changes in some immunological factors and in the distribution of collagen in the regenerated cervical tissue after excision. Moreover, after LLETZ, some cervical features may be subject to modifications, such as the collagen composition and the vaginal microbiota [26]. Finally, although the literature is consistent with a cervical regrowth, little is known about the regeneration of endocervical crypts that may trasverse as far as 5–6 mm from the surface of the cervix before treatment. A reduction in the area of endocervical crypts surface impairs the production of cervical mucus and thus the constitution of the cervical mucus plug, which is the first barrier in preventing ascending bacterial infections during pregnancy. When the mucus plug is smaller or when it is more porous, vaginal bacteria associated with a dysbiosis may infect the amniotic cavity more easily, inducing the inflammatory response associated to PTD [27]. Moreover, women affected by CIN may share the same cervico-vaginal infections as women with PTD [28,29]. A Swedish retrospective population-based register study has shown that not otherwise specified HPV cervical infections shortly before or during pregnancy increase the risk of PTD (*p* = 0.042), and CIN treatment constitutes an additional risk factor for this condition (*p* < 0.001) [30]. Niyibizi et al. have demonstrated that infections from HPV 16 or 18 are associated, independent of any cervical treatment, with an increased risk of PTD [31]. This body of evidence suggests that the increased risk of PTD following LLETZ is linked to multifactorial etiology and not exclusively to cervical biometry as is currently thought.

The main limitation of our study is the small sample size, which did not allow us to perform specific subgroup analyses, such as the comparison of regeneration between patients of childbearing age versus menopausal women or patients who underwent subsequent excisions. For the same reason, we did not consider any correlation with parity, age, oral contraceptive and tobacco use, grade of CIN and cone margins. It should be interesting to explore these variables in a wider study as well as evaluate interpersonal variability of the proposed measurement technique.

In conclusion, our study also confirms literature evidence of cervical regeneration over a period of about 6 months. In our experience, we report a recovery rate of above 90% of both cervical volume and length. We can therefore generically reassure women in childbearing age about starting a pregnancy after the 6th month following excision, although they constitute a group of patients in which it is still necessary to pay attention to the risk of PTD. Unfortunately, a standardized surveillance protocol for fertile women after excision is not available yet, and the LLETZ technique is not standardized but depends on the extension of the transformation zone. Therefore, it could be useful in clinical practice to perform pre- and post-excision cervical analysis, or at least a pre-surgical sonographic evaluation of the cervix (volume and height) and an immediate measurement of the excised cone (volume and height). In this way, we could express the cervical tissue deficit and, consequently, the “potential of cervical regeneration”. When possible, the assessment of the cervical regeneration over time, compared to the baseline values, could be useful in order to achieve a personalized evaluation of the regenerative process. A preventive 3D sonography of the uterine cervix in clinical practice could be also useful to the surgeon in order to more precisely define the upper limit of LLETZ, providing the information that neither bimanual visit nor colposcopy can ensure. The cervix measurement technique that we have proposed has the advantage of identifying an unequivocal reference point in 3D cervical measurement. It allows standardized measurements and comparable prospective evaluations and can be applied with VOCAL™ view with manual contour mode but also with multiplanar 3D mode. We preferred VOCAL™ view to evaluate the cervical volume since, for research purposes, this method appeared theoretically more accurate, although definitely not faster, than the use of the cylinder formula. Nevertheless, our study confirms the very low mean percentage of difference between the two technique and, consequently, the geometric formula of a cylinder from the multiplanar 3D mode dataset can be applied in clinical practice.

## Figures and Tables

**Figure 1 diagnostics-13-00791-f001:**
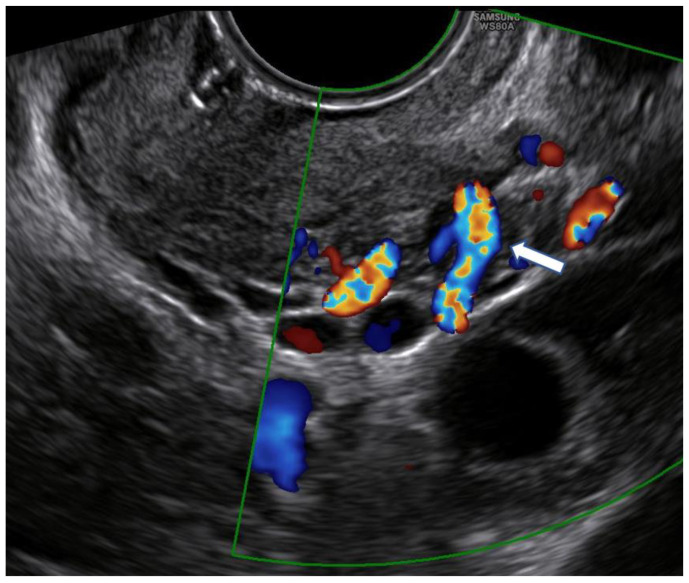
Color Doppler use for the identification of the point where the common trunk of the uterine artery reaches the lateral margin of the uterus (arrow).

**Figure 2 diagnostics-13-00791-f002:**
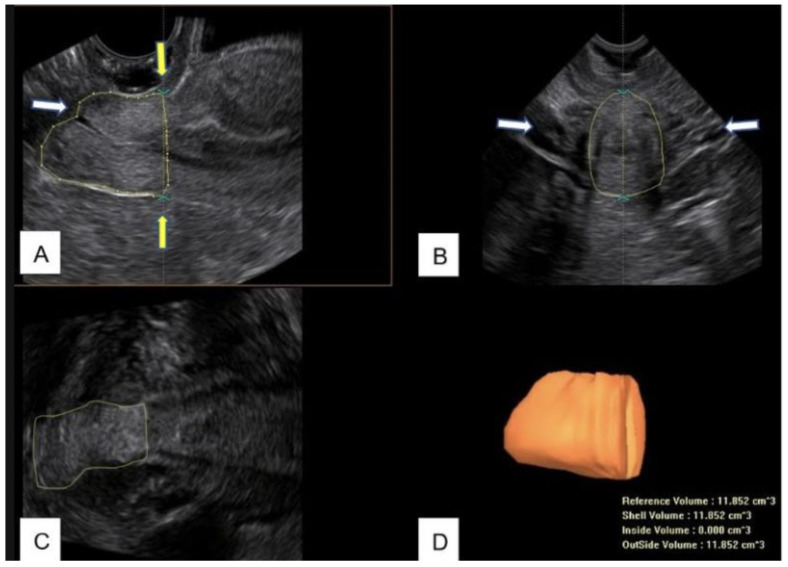
(**A**) B-mode image of the midsagittal view of the cervix after LLETZ. The contour of the cervix is manually drawn to measure the residual volume. The white arrow indicate the uterine external os; the yellow arrows indicate the upper limit of measurement. (**B**) Transverse view of the cervix. The arrows indicate the uterine arteries. (**C**) Coronal view of the cervix. (**D**) Cervical residual volume obtained by Virtual Organ Computer-Aided Analysis (VOCAL™).

**Figure 3 diagnostics-13-00791-f003:**
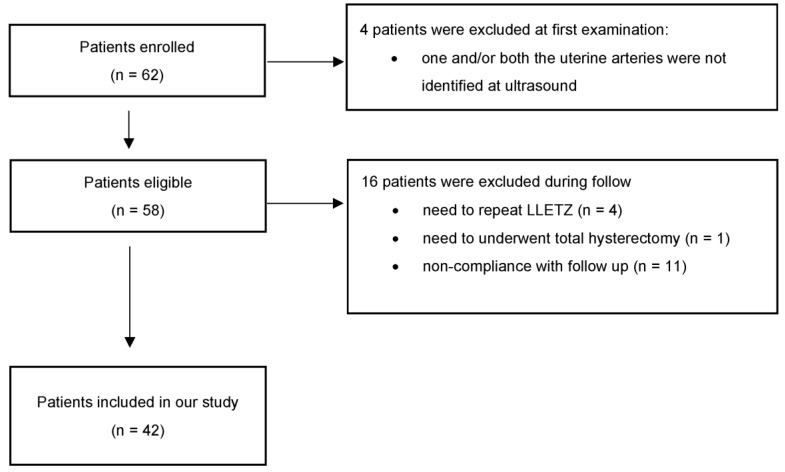
Flowchart with eligible patients and patients included in the study.

**Figure 4 diagnostics-13-00791-f004:**
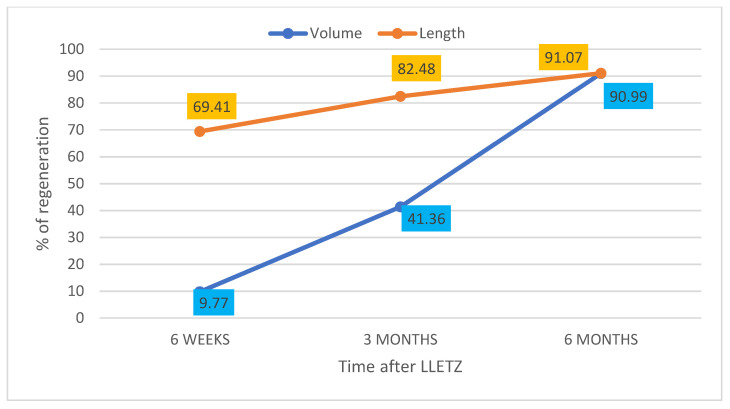
Representation of cervical volume and length regeneration (%) at 6 weeks, 3 months and 6 months compared to the excised cone (tissue deficit regeneration).

**Figure 5 diagnostics-13-00791-f005:**
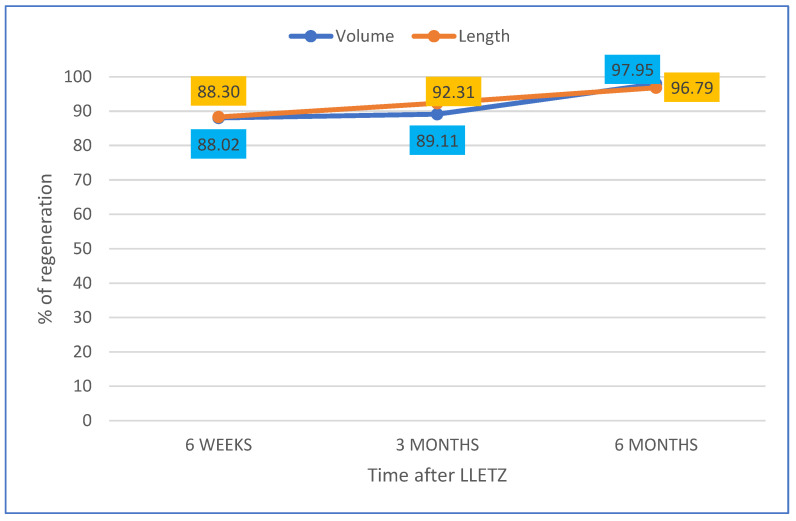
Representation of cervical volume and length regeneration (%) at 6 weeks, 3 months and 6 months not considering the cone size and compared to biometry before LLETZ.

**Figure 6 diagnostics-13-00791-f006:**
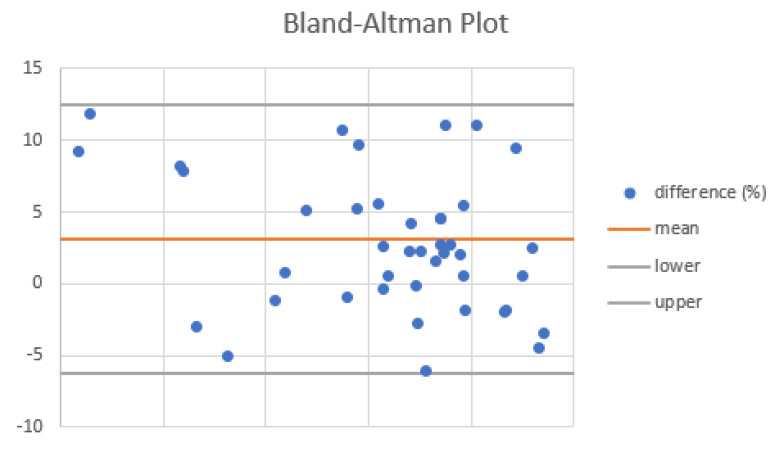
Agreement of VOCAL™ contour mode technique and the geometric formula of the cylinder in the measurement of cervical volume at Bland–Altman plot.

**Table 1 diagnostics-13-00791-t001:** Percentage of cervical growth at follow-up compared to the removed cone size and compared to cervical biometry before excision.

		6 Weeks	3 Months	6 Months
cervical regeneration compared to the removed cone size	volume	9.77	41.36	90.99
	length	69.41	82.48	91.07
cervical regeneration compared to cervical biometry before LLETZ	volume	88.02	89.11	97.95
	length	88.30	92.31	96.79

## Data Availability

All results are reported within the text.

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
