# Peer review of "Ultrasound Control of Cervical Regeneration after Large Loop Excision of the Transformation Zone: Results of an Innovative Measurement Technique"

_diagnostics, 2023, doi:10.3390/diagnostics13040791_

Round 1
Reviewer 1 Report
Dear Authors,
Thank you for allowing me to review your manuscript.
The introduction is well documented and the purpose of the study is clearly expressed.
In the method section should be added information about the Ethical Committee Approval
Table 1 should be added in result section, not in discussion section
At result section some correlations are always a good point.
Discussion and conclusions are well written.
Author Response
Dear Reviewer,
Thank you for your suggestions.
Enclosed you will find our answers.

Reviewer 2 Report
General overview: I was not able to see the red-line of your study. If you aimed to demonstrate the differences between two methods of measurements you need to do this with appropriate design and statistical analysis. If you aimed to evaluate the regeneration according to demographic and clinical factors you need to state this clearly and do this appropriately.
The title of the manuscript is confusing. Please avoid abbreviations in the title and rewrite to be no more than 15 words.
General comments:
- Write the aim of the study at past tense.
- Do not start a sentence with an abbreviation.
- the symbol for decimal is the full stop not comma (2.04 not 2,04).
Abstract:
- It is not clear which pathology is of interest. Furthermore, the context is missing.
- The material and methods information are missing: when and where the study was conducted; which were the inclusion and exclusion criteria; how the measurements were done, by whom and with which device, etc.
- It is not clear how the excision volume was determined.
- The volume seems to be with high dispersion and looks like data did not follow the normal distribution; in this regards means and standard deviation are not appropriate.
- Define abbreviations in the keywords.
Introduction
- define "low-grade lesions "
- "a periodic follow-up is indicated " indicated by whom? define periodic.
- define "simple, low cost and easy to learn procedure "
- define "an increased risk of preterm delivery "
- define RR abréviation.
- provide the 95% confidence interval associated to reported RR.
Materials and methods:
- It is not clear if January 2022 is the month of inclusion or the latest follow-up of the last patient.
- "Patients not available to plan the subsequent follow-ups " not available to plan?
- define "systemic diseases ". Why these patients were excluded?
- The diagnostic criteria are not clear.
- define "skilled sonologist ". Was the monologist blinded?
- briefly describe the applied technique.
- "In previous studies [21], " plural did notify with only one reference.
- define "very accurate and reproducible ".
- "we considered a different " why?
- Use arrows to identify structures in figure 1 and 2.
- "[Song, Papoutsis 2011]. " this reference did not respect the journal requirements.
- "In 6 cases, " this information belongs to the results section. Include here when top hat procedure was considered.
- ""The removed cone was immediately measured" by whom?
- Statistical analysis is superficial written.
Results
- Include here a flowchart with eligible patients and patients included in the study; clearly specify the exclusion.
- 9.77±55.33% provide {min to max} median [Q1 to Q3]
- Put the results baseline and follow-ups in a table to be easily evaluated.
- Figure 3 is distorted.
- Do not duplicate results in text and graphical representations.
- show the agreements between measurements using appropriate statistics (e.g. bland-altman plot; https://www.hindawi.com/journals/cmmm/2019/1891569/).
- "On multivariate analysis, parity, age, oral contraceptive 208 and tobacco use, grade of CIN and cone margins did not affect the regeneration of the 209 cervix. " please provide statistical metrics to support this statement; furthermore, include if appropriate this as secondary outcome of your study.
Discussion
- Begin the discussion by briefly summarizing the main findings.
- Explore possible mechanisms or explanations for these findings.
- Emphasize the new and important aspects of your study.
- Put your findings in the context of the totality of the relevant evidence.
- State the limitations of your study, and explore the implications of your findings for future research and for clinical practice or policy.
- Discuss the influence or association of variables, such as sex and/or gender, on your findings, where appropriate, and the limitations of the data.
- Do not repeat in detail data or other information given in other parts of the manuscript, such as in the Introduction or the Results section.
- This section is not sufficient informative. To say "Many studies have therefore " without a critical evaluation of these studies is just an enumeration is not scientific writing.
- "In an interesting study " such expressions are not appropriate in scientific writing.
- Do not include in this section results (see Table 1).
- Pay attention to duplications.
- "In our study we demonstrated a consistent cervical regeneration over time with both methods. " it is not clear what you want to demonstrate.
Author Response

(The authors gave the same response as above.)

Round 2
Reviewer 2 Report
Improvements had been done on the manuscript, but still I do not see the read-line of your study.
- "Our research firstly considered earlier studies on the cervical regeneration process, with the intent of overcoming previous biases highlighted in literature." Your manuscript is classified as "article". This sentence is not linked with this type of article.
- "Several studies reported progressive cervical tissue regeneration following the excisional procedure, by the sixth post-operative month [6,11-15]." This is an uninformative sentence. Which CIN were evaluated? Which were the main results?
- Define "wide variability".
- It is not clear which is the scientific knowledge (state of the art).
- End the Introduction section with the aim of the study.
- "For this purpose, we used 3D sonography to measure the volume and length of the cervix before LLETZ and, longitudinally, up to six
months after excision, through the application of an innovative and reproducible technique that could overcome measurement bias." - this is methods.
- Define "subsequent follow-ups": how many? when?
- Define "very accurate and reproducible" accurate and reproducible.
- Please use appropriate symbols in the Eq of the volume and "10 x 8 mm".
- Please detail the statistical analysis.
- In Figure 4, please use the full stop as the symbol of decimals.
- Results are duplicated in text and figures.
- Table 1: do not include the "%" symbol in the body of the table.
- It is not appropriate to start the Discussion section with references to the scientific literature.
- Most information in the first paragraph of the Discussion belongs to the state of the art - so it belongs to the Introduction.
Author Response
The answer is in the attachment

Round 3
Reviewer 2 Report
na